# Dynamic Phosphorylation of miRNA Biogenesis Factor HYL1 by MPK3 Involving Nuclear–Cytoplasmic Shuttling and Protein Stability in Arabidopsis

**DOI:** 10.3390/ijms23073787

**Published:** 2022-03-30

**Authors:** Prakash Kumar Bhagat, Deepanjali Verma, Kirti Singh, Raghuram Badmi, Deepika Sharma, Alok Krishna Sinha

**Affiliations:** 1National Institute of Plant Genome Research, Aruna Asaf Ali Marg, New Delhi 110065, India; prakash.k.bhagat@durham.ac.uk (P.K.B.); deepanjali@waksman.rutgers.edu (D.V.); kirtisingh@nipgr.ac.in (K.S.); rbadmi@ucc.ie (R.B.); d.sharma@nipgr.ac.in (D.S.); 2School of Biological and Biomedical Sciences, Durham University, South Road, Durham DH1 3LE, UK; 3Department of Plant Biology, Rutgers, The State University of New Jersey, Piscataway, NJ 08901, USA; 4Raghu Ram Badmi, School of Biological, Earth and Environmental Sciences, University College Cork, T23TK30 Cork, Ireland

**Keywords:** *Arabidopsis thaliana*, HYL1/DRB1, MPK3, post-translational modification, protein stability

## Abstract

MicroRNAs (miRNAs) are one of the prime regulators of gene expression. The recruitment of hyponastic leaves 1 (HYL1), a double-stranded RNA binding protein also termed as DRB1, to the microprocessor complex is crucial for accurate primary-miRNA (pri-miRNA) processing and the accumulation of mature miRNA in *Arabidopsis thaliana*. In the present study, we investigated the role of the MAP kinase-mediated phosphorylation of AtHYL1 and its sub-cellular activity. AtMPK3 specifically phosphorylates AtHYL1 at the evolutionarily conserved serine-42 present at the N-terminal regions and plays an important role in its nuclear–cytosolic shuttling. Additionally, we identified that AtHYL1 is cleaved by trypsin-like proteases into an N-terminal fragment, which renders its subcellular activities. We, for the first time, report that the dimerization of AtHYL1 not only takes place in the nucleus, but also in the cytosol, and the C-terminal of AtHYL1 has a role in regulating its stability, as well as its subcellular localization. AtHYL1 is hyper-phosphorylated in *mpk3* mutants, leading to higher stability and reduced degradation. Our data show that AtMPK3 is a negative regulator of AtHYL1 protein stability and that the AtMPK3-induced phosphorylation of AtHYL1 leads to its protein degradation.

## 1. Introduction

A plant’s development is a complex network of multiple cellular and metabolic pathways which integrate multiple signaling pathways to successfully accomplish its life cycle. One of the evolutionarily conserved signaling pathways in eukaryotes is the mitogen-activated protein kinase (MAPK) cascade. The MAPK cascade is a three-tier kinase cascade where the phosphorylation of MAP kinase kinase kinase (MAPKKK) is activated in response to external/internal stimuli [1]. The activated MAPKKK then phosphorylates the downstream MAP kinase kinase (MAPKK), which eventually phosphorylates the MAP kinase (MAPK). The cytosolic proteins, as well as the transcription factors, are the major targets of activated MAPK phosphorylation [2]. The phosphorylation of the target proteins modulates their subcellular activities in response to stimuli, resulting in plant growth and development. The stress responses are also controlled by the MAPK cascade, where the phosphorylation of target proteins plays a central role in biotic or abiotic stress [3].

MicroRNAs (miRNAs) are endogenous, 21- to 23-nucleotide (nt) small RNAs, which negatively regulate gene expression at the post-transcriptional level [4]. The biogenesis of mature miRNAs is significantly diverse in terms of processing components, as well as the site of processing; however, the functions of the processing factors in both plants and animals are highly conserved [5,6]. In animals, miRNA biogenesis takes place in two cellular compartments: firstly, the primary miRNA transcript synthesized by RNA-polymerase in the nucleus is subsequently processed by two ribonucleases, ribonuclease III Drosha and the double-stranded RNA binding protein, DGCR8/Pasha [7]. After that, the processing of pre-miRNA into mature miRNA/miRNA* is accomplished by another pair of ribonucleases, III dicer and TRBP (HIV TAR RNA-binding protein) [8]. Contrastingly, in plants, only a single pair of ribonucleases, III and DCL1 (dicer-like1), and a double-stranded RNA binding protein (DRB1) known as hyponastic leaves 1 (HYL1) in *Arabidopsis thaliana* are involved in processing pri-miRNA into pre-miRNA, and, finally, into the mature miRNA/miRNA* duplex. The entire biogenesis of miRNA in plants takes place in the nucleus in association with microprocessor accessory proteins, such as serrate (SE) and dawdle (DDL) [5,9].

The post-translational modifications (PTMs) of miRNA biogenesis factors are crucial for proper processing and for miRNA functions in both plants [4,10] and animals [11]. HYL1 is a homologue of DGCR and TRBP, which are regulated by reversible phosphorylation [12]. In animals, ERK1/2 phosphorylates TRBP, which positively regulates miRNA processing by stabilizing the TRBP-containing miRNA microprocessor complex and, thus, its activity [8]. In contrast, the biogenesis of miRNA in plants is negatively regulated by the MAPK (MPK3) and SnRK2 kinases phosphorylation of HYL1 [12,13], whereas dephosphorylation by CPL1/2 and PP4 phosphatases positively regulate biogenesis [14,15]. The phosphorylation of HYL1 has dual functions in modulating miRNA biogenesis by stabilizing the HYL1 protein and inactivating its function [16]. It has been reported that the *mpk3* mutant has a higher level of miRNAs [13], whereas the *snrk2* mutant has reduced miRNA accumulation [13]. These observations suggest that the phosphorylation of HYL1 by these two kinases may have different functions in regulating the HYL1 in miRNA biogenesis and functioning. 

In animals, two distinct dsRBPs, DGCR and TRBP, are present in the nucleus and cytoplasm, respectively [12]. The compartmentalization of miRNA processing and their regulation by phosphorylation is essential for their miRNA-regulated development [11]. HYL1 functions in the nucleus, predominantly in the nuclear dicing body, where both pri-miRNAs and processing factors are localized [14,16,17]. Most recent report have also suggested that HYL1 protects pri-miRNAs from unwanted nuclear exosome activities apart from processing [18]. Another recent study proposed that cytoplasmic HYL1 regulates miRNA-mediated mRNA translation repression in association with argonaute 1 (AGO1) and altered meristem program 1 (AMP1) [19]. Studies have also found that an unknown cytoplasmic protease has been involved in regulating HYL1 proteolysis [16,20]. A recent report suggested that HYL1-cleavage subtilase 1 (HCS1) is a cytoplasmic protease for HYL1 destabilization [21]. They show that the HYL1^K154A^ mutant is insensitive to the proteolytic activity of HCS1, which is also regulated by the light/dark transition [22]. Recently, in plants, the role of reversible phosphorylation is reported with respect to HYL1 reactivation by light to a prolonged dark/shade transition [20,23]. It is also known that CPL1/2 (C-terminal phosphatase-like 1/2) dephosphorylates HYL1 and regulates its functions in miRNA biogenesis [14]. However, the kinase responsible for phosphorylation at any of these sites is still unknown. It is also reported that the phosphorylated form of HLY1 has repressive role during hook opening and that *hyl1-2* knock-out mutants are unable to complete the apical hook formation phase during skotomorphogenic growth [14,24]. HYL1 also regulates the auxin gradient, and the phosphorylated form of HYL1 is an important player in auxin-mediated differential growth during skotomorphogenesis [22].

Additionally, other proteins have also been reported to participate in miRNA biogenesis by affecting HYL1 subcellular activities, including its stability and localization either directly or indirectly [4]. COP1 (constitutive photomorphogenic 1) and SMEK1 (suppressor of MEK1) act as positive regulators of HYL1 protein stability [16,24]. KECTH1 is essential for the nuclear transport of HYL1 from cytoplasm [25]. Another report suggested that DCL1 and HYL1 physically interact with the basic helix-loop-helix (bHLH) transcription factor, phytochrome-interacting factor 4 (PIF4), during the dark-to-red-light transition and regulate miRNA biogenesis in arabidopsis [26]. Based on the above reports, HYL1 is an exciting candidate for further investigation. It will be interesting to know the role of HYL1 phosphorylation on its stability and functioning.

In the present study, we investigated the role of the MAPK-mediated phosphorylation of AtHYL1 and its sub-cellular activity. We report that AtMPK3 phosphorylates AtHYL1 at both the N- and C-terminal regions. We identified that S-42 in the N-terminal region is the site of phosphorylation by MPK3. Further, we report that the phosphorylation of AtHYL1 by AtMPK3 regulates its nuclear–cytosolic shuttling and stability. We also characterized the tryptic nature of cytosolic protease, which destabilizes AtHYL1 by cleaving at the C-terminal regions. Our findings show that AtMPK3 phosphorylates AtHYL1 at S-42 and inhibits its interaction with pri-miRNAs.

## 2. Results

### 2.1. AtHYL1 Is Phosphorylated by AtMPK3

Previously, it has been reported that AtHYL1 is phosphorylated at multiple sites in vivo and that both AtMPK3 and SnRK2 kinases phosphorylate AtHYL1 in vitro [12,13]. To get more insight into AtHYL1 phosphorylation by AtMPK3, we generated N-terminal (1-170 a.a., AtHYL1N) and C-terminal (171-419 a.a., AtHYL1C) constructs of AtHYL1, separating the conserved RNA-binding domains fragment and the disordered C-terminal fragment (Figure 1A) to locate the phosphorylation site(s). An in vitro phosphorylation assay using bacterially expressed AtHYL1N, AtHYL1C, and AtHYL1FL (full-length) indicated that AtMPK3 phosphorylates both N-terminal and C-terminal fragments, as well as the full-length AtHYL1 protein (Figure 1B). To further investigate the interaction specificity, we generated the fragmented constructs by separating the two RNA-binding domains (RBDs) at the N-terminal of AtHYL1 (AtHYL1R1 and AtHYL1R2) (Figure 1A). A yeast two-hybrid assay was conducted to identify the specificity of the protein–protein interaction of the full length and truncated fragments of AtHYL1 (AtHYL1FL, AtHYL1R1, AtHYL1R2, AtHYL1N, and AtHYL1C) with AtMPK3 (Figure 1C and Appendix A). Interestingly, the interaction of AtMPK3 was found with all the analyzed AtHYL1 fragments, including the full-length fragment. The observation also supported our findings from the in vitro phosphorylation assay (Figure 1B and Appendix A). However, when we tested the interaction of different fragments of AtHYL1 with one of its known interacting partners and nuclear protein, serrate (AtSE) [27], using the yeast two-hybrid assay, we found that AtSE specifically interacts only with the N-terminal region and not with the C-terminal region of AtHYL1, which indicates that the N-terminal region ofAtHYL1 is required for the nuclear activity (Figure 1D and Appendix A). Taken together, these observations indicate that AtMPK3 phosphorylates AtHYL1 at multiple sites, in both the N- and C-terminals.

### 2.2. AtHYL1 Is Phosphorylated by AtMPK3 at Serine-42, a Non-Canonical MAPK Phosphorylation Site

To identify the phosphorylation sites on AtHYL1, we performed a protein sequence analysis of AtHYL1 using NetPhos 2.0 (http://www.cbs.dtu.dk/services/NetPhos-2.0/, accessed on 31 March 2017). In silico prediction indicated that there are fifteen putative threonine and seven serine motifs on AtHYL1 as potential phosphorylation sites (Appendix A). Among them, only seven threonine residues were the potential canonical (threonine/serine followed by proline (SP/TP)) MAPK phosphorylation sites. Interestingly, out of these seven threonine residues, only one (T-31) was present at the N-terminal and the rest were present at the C-terminal region (Appendix A). This T-31 residue of AtHYL1 was conserved in *Arabidopsis lyrata* and *Arabis alpine*, indicating that the evolutionary conservation of the putative MAPK phosphorylation site in the species is closely related to *Arabidopsis thaliana* (Appendix A). However, the C-terminal putative phosphorylation sites were partially conserved in *A. alpine* but absent in *A. lyrata*.

We mutated this threionine-31 to alanine (T31 > A) in the AtHYL1N truncated protein and performed the in vitro phosphorylation assay with AtMPK3 (Appendix A). We found that AtHYL1 phosphorylation was not abolished in the AtHYL1N^T31A^ protein, suggesting that threonine-31 is not the phosphorylation site of AtMPK3. We further looked into the AtHYL1 protein sequence to identify any non-conventional MAPK phosphorylation site(s) (proline followed by serine or threonine, i.e., PS/PT). There is a non-conventional MAPK phosphorylation site located at the serine-42 within the N-terminal region, and five serine residues are present at the C-terminal (Appendix A). We mutated S-42 to alanine (AtHYL1N^S42A^) and an in vitro phosphorylation assay was performed (Figure 2A). Mutation in the S-42 completely abolished phosphorylation in AtHYL1N^S42A^, indicating that AtMPK3 phosphorylates AtHYL1 at a non-canonical MAPK phosphorylation site.

To further study the presence of S-42 as a potential MAPK phosphorylation residue, a multiple protein sequence alignment of the AtHYL1 orthologues from different species was carried out. The alignment data revealed two interesting insights (Appendix A): first, the N-terminal domains are highly conserved in all analyzed species, which is not so for the extended C-terminal regions. Secondly, the S-42 residue of AtHYL1 phosphorylated by AtMPK3 is evolutionarily conserved in most of the species. The maximum amino acid similarities were observed between the arabidopsis and brassica members (Appendix A). In brassica, the C-terminal repetitive regions of 28 a.a. were absent, and even the multiple putative phosphorylation threonine residue was absent, indicating the possible selection pressure during evolution (Appendix A). Taken together, this observation is in agreement with the conserved interactions between AtHYL1/AtDRB1 and AtMPK3 in arabidopsis [12].

To further investigate the role of AtHYL1 phosphorylation at dsRBD1 and S-42 by AtMPK3, we took help from the AtHYL1 orthologues present in rice [13]. It was observed that in rice, one of the OsDRB paralogues, OsDRB1-4, lacks this conserved serine motif in its dsRBD1 (Figure 2B), where serine is replaced by histidine (PS < EH). We exploited this naturally occurring mutation to investigate the potential phosphorylation site in the dsRBD1 of rice by OsMPK3. We generated the N- and C-terminal fragmented constructs of OsDRB1-1, OsDRB1-2, and OsDRB1-4 for bacterial expression and used the fragmented proteins (Figure 2C) in an in vitro phosphorylation assay by OsMPK3 (rice MPK3). Excitingly, we observed the phosphorylation in both the N- and C-terminal fragments of OsDRB1-1 (Figure 2D) and OsDRB1-2 (Figure 2E), and only in the C-terminal fragment of OsDRB1-4 but not in the N-terminal fragment of OsDRB1-4 (Figure 2F). These observations, combined with the yeast two-hybrid results using fragmented AtHYL1 (Figure 1B,C), point out that the S-42 in the N-terminal dsRBD1 is the MPK3 phosphorylation site conserved in distantly related species. Taken together, these data indicate that HYL1 is phosphorylated at the serine-42 position in the N-terminal region.

### 2.3. Mutation in AtMPK3 Leads to the Hyper-Phosphorylation of AtHYL1

Previous reports suggested that the phosphorylation of AtHYL1 inhibits its interaction with the pri-miRNA transcript [14,16]. To find the role of AtMPK3 in AtHYL1 function in vivo, we analyzed its protein abundance in *mpk3* mutant seedlings. Surprisingly, we found that AtHYL1 migration was slower, and its band was at a higher molecular size, compared to Col-0 (Figure 3A). The slow migration of AtHYL1 in the *mpk3* background suggests an additional unknown post-translational modification(s). AtHYL1 is a phospho-protein and exists in two isoforms: slow migrating phosphorylated and fast-migrating dephosphorylated AtHYL1 [14]. Thus, we treated the total protein from the *mpk3* mutant with calf intestine alkaline phosphatase (CIAP) and performed an immunoblotting of the dephosphorylated protein (Figure 3B). Interestingly, it was observed that in the presence of phosphatase, the mobility of the protein was significantly restored, suggesting that AtHYL1 is present in a hyper-phosphorylated state in the *mpk3* background. To further confirm the above result, we treated the Col-0 seedlings with or without protein phosphatase inhibitors (β-glycerophosphate, sodium fluoride, and sodium orthovanadate) overnight, and total protein was extracted in the extraction buffer containing these inhibitors, followed by immunoblotting with an anti-HYL1 antibody (Figure 3C). Only a single AtHYL1 band was observed in the lane where no inhibitor was used. However, a higher molecular weight AtHYL1 was observed, along with a lower band in the inhibitor-treated seedlings (Figure 3C). These data suggest that the inhibition of phosphatases activity may enhance the accumulation of phosphorylated AtHYL1 in the cell. Additionally, the accumulation of phosphorylated AtHYL1 as observed in *mpk3* mutant background might be due to the low activity of phosphatases or higher kinase activity by other kinases.

### 2.4. AtMPK3 Promotes AtHYL1 Protein Degradation

Previous reports suggest that phosphorylation status at S-42 does not affect AtHYL1 stability; rather, it inhibits the interaction with pri-miRNAs and the stability of AtHYL1 is enhanced by phosphorylation at S-159 by unknown kinases [16]. To find out if AtHYL1 phosphorylation at S-42 by AtMPK3 affects its stability, we treated Col-0 seedlings with MAPK inhibitors overnight, and then subjected them to immunoblotting. We found that the inhibitor treatment enhanced AtHYL1 protein accumulation as compared to the untreated samples, suggesting that phosphorylation by MAPK regulates its protein stability (Figure 4A). The result indicates that a higher AtHYL1 protein stability in the *mpk3* mutant (Figure 3) is due to a lack of AtMPK3-induced phosphorylation in the protein. 

To further explore the role of reversible phosphorylation in AtHYL1 homeostasis, we performed a cell-free degradation assay using bacterially purified GST-AtHYL1 protein and incubated it with the total protein extract from the Col-0 seedlings. Immunoblotting with anti-GST indicated that AtHYL1 is degraded by proteolytic cleavage and results in the accumulation of a GST-tagged, N-terminal degraded products with a decrease in the full length of AtHYL1 level (Figure 4B). To confirm the role of AtMPK3 in AtHYL1 degradation, we performed a similar experiment with the total protein extract from the *mpk3* mutant seedlings (Figure 4B). We found that the degradation of AtHYL1 is faster in *mpk3* than it is in Col-0. These data suggest that dephosphorylated AtHYL1 is a short-lived protein that rapidly degrades through proteolysis. 

To confirm the role of AtHYL1 phosphorylation in vivo, the total protein isolated from Col-0 and *mpk3* was used to assay the stability of endogenous AtHYL1 by incubating it for different time points. In Col-0, we found that the degradation of AtHYL1 was faster and diminished at 60 min (Figure 4C). In contrast, AtHYL1 degradation was delayed up to 60 min and two isoforms of AtHYL1 could be clearly seen with the incubation time in the *mpk3* total extract (Figure 4C). The presence of two isoforms were previously identified as phosphorylated and dephosphorylated AtHYL1, where dephosphorylation is a pre-requisite for its degradation and phosphorylated AtHYL1 is nuclear-localized and is a stable pool [16]. Taken together, these data indicate that dephosphorylated AtHYL1, as seen in the bacterially purified protein, is more prone to degradation. In contrast, hyperphosphorylated AtHYL1, as seen in the *mpk3* extract, is more stable and its degradation rate is slower due to the additional step required for the dephosphorylation of AtHYL1, which is a pre-requisite for its degradation. 

However, the stability of AtHYL1 is reported to be tightly regulated by prolonged darkness and the dark-to-light transition [16,19]. Moreover, it is also reported that in mutants like *cop1*, *cpl1, cpl2,* and *smek1,* AtHYL1 is drastically reduced along with the reduction in miRNAs, and thus they are considered as positive regulator of AtHYL1 [15,16,23,24]. Further, a prolonged darkness (more than 24 h) degrades non-phosphorylated forms of AtHYL1 more rapidly than phosphorylated forms [16]. To confirm the role of AtMPK3 in the dark-dependent degradation of AtHYL1, we analyzed the stability of the AtHYL1 protein in the Col-0 and *mpk3* mutant by keeping the plants in light and dark conditions for 24 and 48 h. It was observed that in *mpk3* mutants, there was no visible degradation of AtHYL1 in the dark, and it also revealed hyper-phosphorylated forms under both the treatments (Figure 4D). In contrast, a slight degradation of AtHYL1 was observed in Col-0. The prolonged darkness drastically reduced the AtHYL1 protein abundance in both the Col-0 and *mpk3* plants; however, even on longer exposure, the presence of protein can be seen in *mpk3* mutants (Figure 4D). Interestingly, it was noticed that only the hyper-phosphorylated form of AtHYL1 was stable in the *mpk3* background in prolonged darkness, which is consistent with the previous report where only phosphorylated AtHYL1 is partially resistant to dark-induced degradation [16].

### 2.5. The Nuclear–Cytosolic Localization of AtHYL1 Is Strictly Regulated by Cytosolic Proteolysis

To elucidate the role of HYL1 proteolysis and its subcellular activities that might be regulated by post-translational modification by AtMPK3, we tested the in vitro degradation of the protein using a universal protease, trypsin. We have shown above that bacterially expressed AtHYL1 proteins undergo proteolysis in vitro when incubated with plant extracts (Figure 4D). We further incubated the bacterially purified HIS-AtHYL1 with trypsin for the indicated time period. It was observed that AtHYL1 protein is cleaved into a 26 kDa N-terminal fragment (Figure 5A). Next, we challenged HIS-AtHYL1N protein with trypsin to check its stability. Interestingly, the N-terminal of HYL1 was unaffected by the proteolytic cleavage as compared to the full length AtHYL1 (Figure 5B). This analysis supported the recent report where HCS1 protease is confirmed as the protease for HYL1-destabilization [20] that might have an action like that of trypsin proteases. The incubation of bacterially purified full length AtHYL1 and AtHYL1N with crude protein extracted as a source of proteases from Col-0 further proves the tryptic nature of these proteases. Additionally, immunoblotting with anti-HYL1 antibody reacts with both full length AtHYL1 and its proteolytic product, but not with AtHYL1N (Figure 5B). This observation suggests that the trypsin proteolysis generated a mixture of both N- and C-terminal products, and the antibody used here has been generated against the C-terminal peptide. The amino acid analysis of AtHYL1 further suggests the presence of multiple trypsin cleavage sites at the highly disordered C-terminal region (Appendix A).

Next, we asked whether AtHYL1 proteolysis has any role in modulating its subcellular localization. We hypothesized that if AtHYL1undergoes proteolysis, then its subcellular localization pattern may vary depending upon the presence of a fluorescence tag at the amino- and carboxyl-terminals of AtHYL1. We first tagged the green fluorescence protein (GFP) to the amino-terminal of the full length AtHYL1, and its localization was studied by transient overexpression in *N. benthamiana* leaves (Figure 5C). The GFP-AtHYL1 was found to be localized in both the nucleus and cytoplasm. Next, we tagged full-length AtHYL1 with GFP at its carboxyl-terminal and localization was analyzed (Figure 5D). Interestingly, the localization of AtHYL1-GFP was found solely in the nucleus. The contrasting cytosolic localization of full-length AtHYL1 by the mere presence of GFP at two distinct terminals, combined with a cell-free degradation assay, further suggest that AtHYL1 proteins undergo cytoplasmic degradation resulting in the production of the 26kDa N-terminal, which is stable as compared to that of the highly unstable C-terminal product. Taken together, these data indicate that AtHYL1 is a short-lived protein whose subcellular activity is subjected to multiple regulatory steps, such as proteolysis, phosphorylation, and dephosphotylation, to fine-tune miRNA biogenesis.

### 2.6. Co-Expression of AtMPK3 Regulates AtHYL1 Subcellular Localization

To investigate the role of MPK3 in AtHYL1 localization, we used the N-terminal-tagged fluorophore GFP-AtHYL1, which is localized at both the cytoplasm and in the nucleus, in contrast to the C-terminal-tagged fluorophore, AtHYL1-GFP, which can be easily monitored any additional effect of AtMPK3-induced phosphorylation. The BiFC assay suggested that both AtHYL1 and AtMPK3 interact in the nucleus, indicating that AtHYL1 might be phosphorylated in the nucleus (Appendix A). As observed above, GFP-AtHYL1 alone showed cytoplasmic and nuclear localization, but the co-expression of AtMPK3-HA restricted the GFP fluorescence to only the nucleus (Figure 5E). These data lead us to conclude that the phosphorylation of AtHYL1 by AtMPK3 regulates its subcellular localization. 

The above observations suggest that AtHYL1 is a nucleo–cytosolic protein, and all the interactions of AtHYL1 have been reported mainly in the nucleus and, more precisely, in the dicing bodies [23]. To get a better insight, we employed a BiFC assay where the YFP amino- and carboxyl-half fluorophores were tagged at either the N-terminal (n/cYFP-AtHYL1) or the C-terminal (AtHYL1- n/cYFP) of full-length AtHYL1. We found that the transient co-expression of AtHYL1-nYFP and AtHYL1-cYFP in tobacco leaves showed nuclear dimerization, as reported previously (Appendix A) [23]. In contrast, AtHYL1 predominately formed dimers in the nucleus, also with a weak cytosolic interaction with nYFP-AtHYL1 and cYFP -AtHYL1 in tobacco leaves (Figure 5F). This data suggests that intact AtHYL1 predominantly forms dimers in the nucleus and the weak YFP signal observed in the cytosol might be due to the N-terminal domains as a result of cytoplasmic proteolysis. 

## 3. Discussion

### 3.1. AtMPK3 Phosphorylates AtHYL1 at the Serine-42 Position

AtHYL1, a core miRNA biogenesis factor, has recently attracted lot of attention, and there have been ever-increasing reports of new players, hypotheses, and interactors of this important protein [14,15,16,19]. Here, we present new insights on the role of a MAPK, AtMPK3, in regulating the subcellular activities of AtHYL1. We reported earlier that AtHYL1 is a phosphorylation target of AtMPK3 [12]. In the current report, we primarily focused our studies on the phosphorylation of AtHYL1 by AtMPK3 and phosphorylation-induced AtHYL1 subcellular functions. We found that AtMPK3 interacted with both the N- and C-terminals of AtHYL1 in a Y2H assay (Figure 1C,D). Consequently, an in vitro phosphorylation assay using bacterially purified full-length, fragmented N- and C-terminals of AtHYL1 showed a strong phosphorylation by AtMPK3 (Figure 1B). A protein sequence analysis of AtHYL1 suggested that it has multiple phosphorylation sites, as depicted by the NetPhos server, including the previously reported seven serine residues located at the N-terminal [14]. Additionally, there are seven potential MAPK phosphorylation threonine sites, out of which only one, T-31, was present at the N-terminal and found to be evolutionarily conserved, while the other six were present at the C-terminal region. However, an in vitro phosphorylation assay using AtHYL1N^T31A^ still showed phosphorylation eliminating its candidacy as a phosphorylation site for AtMPK3. Further, we identified an S-42 amino acid located at the N-terminal as a non-conventional MAPK phosphorylation site, which was evolutionarily conserved in all other plants. An in vitro kinase assay confirmed that the substitution of S-42 to alanine abolished the phosphorylation of AtHYL1N. The S-42 was also previously identified to be phosphorylated in vivo; however, the kinase that phosphorylates at the S-42 site was unknown. Therefore, we concluded that AtMPK3 phosphorylates the S-42 of AtHYL1 (Figure 2A). Interestingly, the arabidopsis AtHYL1 orthologue in rice, OsDRB1, has four orthologues: OsDRB1-1, OsDRB1-2, OsDRB1-3, and OsDRB1-4. Out of these four, OsDRB1-2 and OsDRB1-3 have codes for the same gene and are the result of endo-duplication [12]. Protein sequence analysis revealed that the S-42 residue of AtHYL1 was evolutionarily conserved in both OsDRB1-1 and OsDRB1-2; however, it was absent in OsDRB1-4. Our in vitro phosphorylation assay by OsMPK3 revealed that the N-terminals in OsDRB1-1 and OsDRB1-2 were phosphorylated, but the N-terminal of the OsDRB1-4 (Figure 2D–F) was not. Based on these results, we concluded that the evolutionarily conserved S-42 of AtHYL1 is a site of phosphorylation by AtMPK3. However, the functional relevance of AtHYL1 phosphorylation at the C-terminal regions by AtMPK3 cannot be ignored.

### 3.2. AtMPK3 Negatively Regulates AtHYL1 Protein Accumulation 

The phosphorylation of AtHYL1 has been shown to regulate its subcellular localization and stability [16]. However, S-42 phosphorylation does not affect its subcellular localization, but abolishes the interaction with pri-miRNAs [16]. Therefore, we established that AtHYL1 phosphorylation at S-42 by AtMPK3 negatively regulates its interaction with pri-miRNAs in planta. We observed that AtHYL1 protein mobility was slower in the *mpk3* mutant due to a higher accumulation of phosphorylated AtHYL1 by other kinases (Figure 3A). The treatment of CIAP of total crude lysate from *mpk3* restored its mobility, suggesting the presence of phosphorylated AtHYL1 (Figure 3B,C). A previous study also supports this observation that phosphorylation stabilizes AtHYL1 within the nuclear bodies as an inactive AtHYL1 pool [16]. We also found that the inhibition of MAPK signaling using the PD98059 MAPK inhibitor stabilized AtHYL1 proteins in WT (Figure 4A). Thus, we can state that AtHYL1 stability is negatively regulated by AtMPK3, but the accumulation of phosphorylated AtHYL1 in the *mpk3* mutant further suggests a possible antagonistic role with phosphorylation at other sites of AtHYL1, leading to its higher stability. 

### 3.3. AtHYL1 Exists in a Hyper-Phosphorylated State in the mpk3 Mutant Background 

AtHYL1 stability is known to be regulated by phosphorylation at the S-159 residue [16]. Here, we showed that AtMPK3 phosphorylates the S-42 residue at the N-terminal of AtHYL1, which was not reported to have any role in maintaining its stability. However, we cannot rule out the role of the C-terminal phosphorylation of AtHYL1 by AtMPK3, which we also showed be phosphorylated. Based on the above observations, we propose two possible reasons for the higher stability of AtHYL1 in the *mpk3* mutants. First, it may be that AtMPK3-induced phosphorylation destabilizes the AtHYL1 protein having an antagonistic role with S-159 kinase(s). Secondly, the activity of the phosphatase responsible for the dephosphorylation at S-159, or another site apart from S-42, is low, resulting in a higher accumulation of phosphorylated AtHYL1 in the *mpk3* background. 

Further, it has been shown that dark or shade conditions induce AtHYL1 degradation by the dephosphorylation of AtHYL1, but in dark-to-light transitions, the phosphorylated AtHYL1 is quickly dephosphorylated to activate the miRNA biogenesis [16]. We showed that light-to-dark-induced AtHYL1 degradation is compromised in the *mpk3* mutants. This may be due to the additional dephosphorylation steps required in the *mpk3* background prior to AtHYL1 degradation [15,16]. This observation was further clarified by using bacterially purified AtHYL1 proteins in a cell-free degradation assay where dephosphorylated AtHYL1 was seen to be more susceptible to degradation as compared to that of endogenous AtHYL1 protein in the *mpk3* mutant background (Figure 4), suggesting that the additional post-translational modification is crucial for its stability. 

### 3.4. Subcellular Localisation of AtHYL1 Is Tightly Governed by Cytoplasmic Proteases 

It was shown that the degradation of AtHYL1 is tightly regulated by proteolysis at cytosol by an unknown protease [19]. So far, the identity or activity of cytosolic protease has not been determined. Using an in vitro degradation assay, we found that the tryptic nature of the crude lysate exhibited the similar proteolytic degradation product as seen in the trypsin-treated AtHYL1 protein (Figure 5A). We further found that the C-terminal of AtHYL1 has multiple tryptic cleavage sites as compared to the N-terminal, and cleavage at any of these sites might regulate its stability and also other subcellular activities, such as localization. However, previous reports suggest that the C-terminal of AtHYL1 has no evolutionary role in its function in miRNA biogenesis, and hence it is largely ignored [28]. Here, we found that the C-terminal of AtHYL1 is partially important for its subcellular functions. Using a bacterially purified AtHYL1N truncated protein, we observed its higher stability in a cell-free degradation assay and in trypsin treatment as compared to full-length AtHYL1, which has C-terminal regions (Figure 5A,B). These data suggest that AtHYL1 is destabilized by the action of cytosolic protease at the C-terminal cleavage sites that also reside in the nuclear localization sequence. 

### 3.5. AtHYL1 Is Localised to Both Nucleus and Cytoplasm 

We further showed the importance of the C-terminal regions in the subcellular localization by tagging the GFP at either the N- or C-terminals of full-length AtHYL1. The GFP-AtHYL1 protein was localized to the nucleus, as well as in the cytoplasm (especially in the periphery of cell), which was consistent with the previous reports [14,16]. However, when GFP was tagged to the C-terminal (AtHYL1-GFP), it showed only nuclear localization and no cytosolic localization was observed, as was seen when GFP was tagged to the N-terminal (GFP-AtHYL1) (Figure 5C,D). We also excluded the possibility of tags on any of the protein terminals which may exhibit the contrasting localization patterns, as a similar observation was reported by Fang and Spector in 2007 [23]. Based on the cell-free degradation assay and AtHYL1 localization in vivo, we propose that intact AtHYL1 is mostly present in the nucleus; however, upon protease action, the 26 kDa proteolytic N-terminal products can be seen in the cytosol. Thus, the C-terminal of AtHYL1 has a role in regulating its stability, as well as its subcellular localization. It was reported earlier that the AtHYL1 C-terminal is partially important for pri-miRNA interaction, whereas full-length AtHYL1 has a higher binding affinity compared to that of N-terminals or double-stranded RNA binding domains alone [29]. Therefore, cleavage at the C-terminal of AtHYL1 may hamper its nuclear accumulation, as well as its affinity for pri-miRNAs. 

There are reports which suggest that phosphorylation plays an important role in protein turnover [28,30,31]. The phosphorylation of the WRKY-type transcription factor in *Coptis japonica* regulates its translocation to cytosol and its subsequent degradation by cytosolic/vascular proteases [28]. Additionally, multisite phosphorylation has been proposed to regulate an effective switching-like protein degradation [28]. Similarly, AtHYL1 forms dimers in the nucleus, where it also interacts with other miRNA biogenesis factors [23,32,33,34]. Here, we also showed that localization of GFP-AtHYL1 was restricted to the nucleus when co-expressed with AtMPK3 (Figure 5E). The result that AtHYL1 forms a weak dimer in the cytoplasm (Figure 5F) suggests that it may have other functions. However, the biological function of this interaction remains to be explored. Overall, our observation adds another function of AtHYL1 and provides new insights to the existing knowledge of AtHYL1 functioning and regulation. 

## 4. Materials and Methods

### 4.1. Plant Growth Conditions

*Arabidopsis thaliana* (Col-0) and rice (*Oryza sativa L. indica* cultivar group var. pusa basmati 1) were used in the study. Arabidopsis seeds were surface-sterilized with 2% sodium hypochlorite and 0.02% Triton X-100 solution, plated on ½ Murashige and Skoog (MS) media, and incubated in the dark at 4 °C. After 3 days of stratification, the plates were transferred to 22 °C under a long day 16/8 light-dark cycle or constant light (80 μmolm-2 s-1, warm white-830/cool white-840, Philips, Amsterdam, The Netherlands) conditions. Rice plants (*Oryza sativa L. indica* cultivar group var. pusa basmati 1) were grown either in growth chambers (Scilab instrument, Taiwan, China) or in a greenhouse at 28 °C with a 16/8 light-dark cycle. Seedlings 7–10 days old were harvested and used for RNA isolation and cDNA preparation.

### 4.2. Yeast Two-Hybrid Assay 

The full-length CDS of *AtMPK3*, *AtSE*, and *AtHYL1*, along with the deletion fragments, were amplified from 7–10 day-old arabidopsis cDNA samples using the appropriate primers (Appendix A) by Phusion DNA-polymerase (NEB, Ipswich, MA, USA) and cloned in-frame in pGADT7 (AD) and pGBKT7 (BD) vectors (Clontech, CA, USA). Protein–protein interactions were performed according to the manufacturer’s protocols and as mentioned in a previous report [12]. 

### 4.3. In Vitro Phosphorylation Assay

To perform the in vitro phosphorylation assay, full-length CDS of *AtMPK3*, *OsMPK3*, and *OsDRB1-1* were cloned in–frame in pGEXT42 protein expression vector (Appendix A), whereas *AtHYL1*, *OsDRB1-2,* and *OsDRB1-4* full-length and deletion constructs were cloned in the pET series vectors (pET21c/pET28a). The constructs were transformed in *E. coli BL21* competent cells and the gene expression was induced by the addition of 1 mM IPTG in culture media. His- and GST-tag proteins were purified using the Ni-NTA (Qiagen, Hilden, Germany) and GST- beads (GE Healthcare Bio-sciences, Uppsala, Sweden), respectively. The in vitro phosphorylation assay was performed as previously described [12].

### 4.4. In Vitro Protease Sensitivity Assay

The protease sensitivity assay using bacterially purified AtHYL1FL and AtHYL1N terminal was performed according to previous reports [35,36]. The samples were loaded on 12–15% SDS-PAGE, and the protein bands were visualized using Coomassie Brilliant Blue (CBB) staining. The Western blotting of the same experiment setup was further used to analyze by anti-AtHYL1 antibody (5:10,000) (AS06136, Agrisera, Vannas, Sweden), according to the report described earlier [37].

### 4.5. In Vitro Cell-Free Degradation Assay

The in vitro protein degradation assay was performed as previously described [19], with minor modifications. Briefly, 2–5 µg of bacterially purified His-tagged AtHYL1, AtHYL1N, and GST-AtHYL1 proteins were incubated with the total crude protein extract (10 µg) from Col-0 and *mpk3* for the indicated time periods at 30 °C. The samples were harvested and the reaction was stopped by adding SDS-sample loading dye, heating at 95 °C for 5 min, and then loading samples on 12–15% SDS-PAGE gel. Immunoblotting was performed with either anti-AtHYL1 or ant-GST antibody. For the endogenous AtHYL1 protein degradation assay, 30 to 40 µg of total crude extracts were incubated for the indicated times and immunoblotting was performed as above.

### 4.6. Protein Subcellular Localization and Bimolecular Complementation (BiFc) Analysis

For localization studies, AtHYL1FL was cloned in the Gateway binary vectors pGWB5 and pGWB6 with superfold-green fluorescence protein (sGFP) tags at the carboxyl- and amino-terminals of HYL1, respectively. The recombinant pGWB5 and pGWB6 (encoding HYL1-GFP and GFP-HYL1) were finally transformed into *Agrobacterium* GV3101. BiFc analysis using AtHYL1-AtMPK3 and localization by agro-infiltration were performed according to a previous report [12].

### 4.7. In Vitro Phosphatase Treatment

Total proteins were extracted from light-grown *mpk3* seedlings with protein extraction buffer (100 mM Tris·HCl (pH 6.8), 20% glycerol, 20 mM dithiothreitol, 1 mM PMSF, 1 × protease inhibitor, and 100 μM MG132) and 30–40 µg total soluble proteins were incubated with calf intestinal alkaline phosphatase (CIAP) according to the manufacturer’s protocol at 37 °C, followed by immunoblotting with anti-AtHYL1 antibody as described above.

### 4.8. Light-to-Dark Transition Assay

The light-to-dark-induced AtHYL1 degradation was monitored as in a previously described procedure [16] using Col-0 and *mpk3* seedlings. The AtHYL1 and AtMPK3 proteins were detected by anti-HYL1 and anti-MPK3 antibody.

### 4.9. Evolutionary Relationship of AtHYL1 with DRB1 from Other Plants

The protein sequences were retrieved from the Uniprot database by a BLAST search using the AtHYL1 protein sequence as an input. The phylogenetic tree was constructed using the MEGA7 program for alignment of the sequences and construction of the phylogenetic tree by using a maximum likelihood method based on the JTT matrix-based model. The tree with the highest log likelihood (−3606.13) is shown. Initial tree(s) for the heuristic search were obtained automatically by applying the Neighbor-Join and BioNJ algorithms to a matrix of pairwise distances estimated using a JTT model, and then selecting the topology with the superior log likelihood value. The tree was drawn to scale, with branch lengths measured in the number of substitutions per site. The analysis involved 14 amino acid sequences. All positions containing gaps and missing data were eliminated. There was a total of 244 positions in the final dataset. Evolutionary analyses were conducted in MEGA7.

### 4.10. Multiple Protein Alignment Analysis

The evolutionary conservation of dsRNA binding domains and putative MAPK sites were analyzed between the dsRNA binding proteins (DRB1/HYL1) from *Arabidopsis thaliana* and other monocots, as well as dicots. For this, protein sequences were downloaded from the National Centre for Biotechnology Information (NCBI) and Uniprot. Multiple protein sequences were aligned by the Uniprot align tool. Initially, we aligned the HYL1 protein sequence from *Arabidopsis thaliana* with *Arabidopsis lyrata* (UniprotKB Identifier D7KJT2), *Arabis alpine* (A0A087HMB7), and with other close relative brassica members (*Brassica napus*/Q5IZK5, *Brassica oleracea*/A0A0D3DNR2, and *Brassica_rapa*/M4EPS2). Later, to further analyse the evolutionary basis of the conservation of amino acid residues, we performed multiple sequence alignment with a few monocots (*Populustrichocarpa*/B9H6U2, *Vitisvinifera*/A5BNI8-1, and *Solanumlycopersicum*/K4BU80) and dicots (*Zea mays*/B6TPY5-1, *Setariaitalica*/K3Y7A9, *Oryza sativa* subsp. *indica*/I2DBG3, *Oryza sativa* subsp. *japonica*/Q0IQN6, *Brachypodiumdistachyon*/A0A0Q3F254, and *Musa acuminate*/M0RRC4).

## 5. Conclusions

In this study, we propose that AtMPK3 is crucial in regulating and maintaining AtHYL1 protein abundance and nucleo–cytosolic shuttling. However, our findings also open up a new area of research on AtHYL1 and its post-translational modifications in the absence of AtMPK3. Thus, based on our results, we propose following three questions that need to be investigated further: (i) what is the role of C-terminal regions in AtHYL1 function; (ii) what is the role of AtHYL1 and/or N-terminal products in the cytosol, apart from its degradation; and (iii) is the antagonistic role of AtMPK3 with that of unknown kinase(s) responsible for the hyper-phosphorylation of AtHYL1 in the absence of AtMPK3.

## Figures and Tables

**Figure 1 ijms-23-03787-f001:**
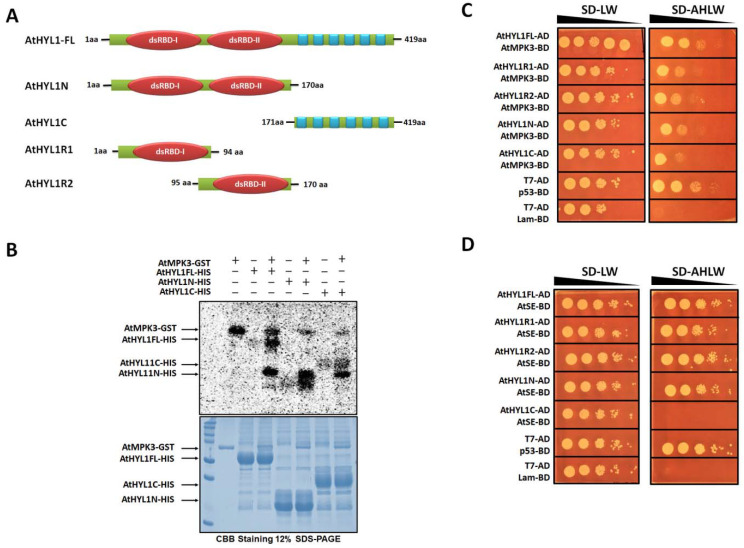
AtMPK3 phosphorylates at both terminals of AtHYL1. (**A**) A diagrammatic representation of AtHYL1 protein and its domain architecture. The N-terminal and C-terminal half used for protein expression, in vitro experiments, and Y2H are indicated by amino acid numbers. (**B**) An in vitro phosphorylation assay showing the phosphorylation of HYL1 full length (AtHYL1FL) and N-terminal (AtHYL1N) and C-terminal (AtHYL1C) regions by AtMPK3. The plus and minus signs represent the presence and absence of proteins. (**C**,**D**) A yeast two-hybrid assay showing the interaction of different versions of AtHYL1 (AtHYL1FL, AtHYL1N, AtHYL1C, AtHYL1R1—first dsRBD, and AtHYL1R2—second dsRBD) with AtMPK3 and AtSE, respectively. The images were taken after growing the yeast on respective medium at 28 °C for 3 days. All the experiments were performed three times to validate the results.

**Figure 2 ijms-23-03787-f002:**
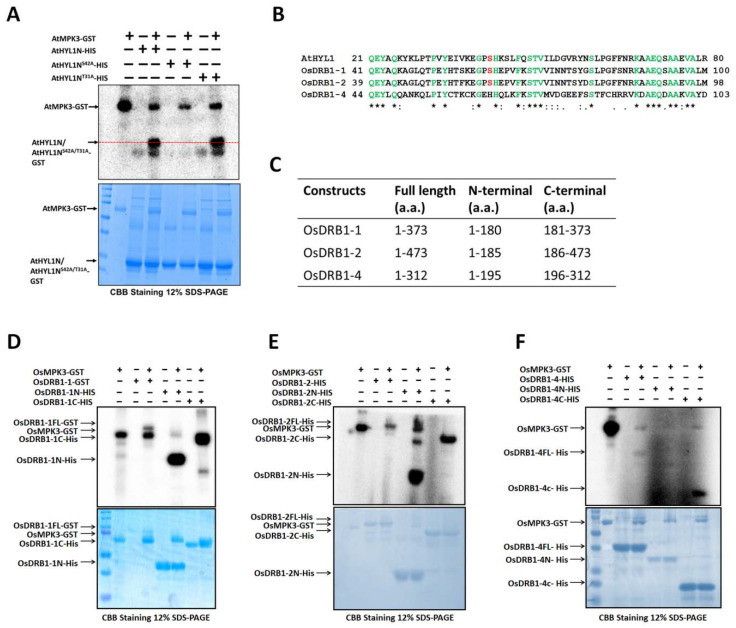
AtMPK3 phosphorylates AtHYL1 at the serine-42 residue at the N-terminal domain. (**A**) The in vitro phosphorylation of wild type and phosphor-null AtHYL1N by AtMPK3. Two putative MAPK phosphorylation sites on AtHYL1 (threonine-31 and serine-42) were mutated individually to alanine (AtHYL1N^T31A^ and AtHYL1N^S42A^) and wild type AtHYL1N was incubated with AtMPK3 in a kinase buffer. +, − signs indicate presence and absence of respective proteins. (**B**) The amino acid alignments of AtHYL1 and OsDRBs showing the conserved serine residue marked with a red box, which is absent in OsDRB1-4. (*, **, ****) represents conserved amino acids. (**C**) The amino acid length of N and C-terminals of OsDRBs used for protein expression. An in vitro phosphorylation assay showing the phosphorylation of full-length and truncated proteins of OsDRB1-1 (**D**), OsDRB1-2 (**E**), and OsDRB1-4 (**F**) by OsMPK3. The upper images are the autoradiographs and the lower are CBB stainings of the respective gels.

**Figure 3 ijms-23-03787-f003:**
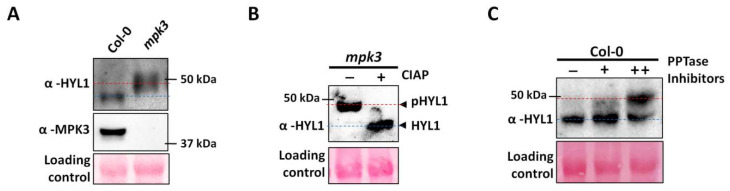
MPK3 is a negative regulator of HYL1 protein abundance. (**A**) *mpk3* mutants accumulate hyper-phosphorylated HYL1. Total proteins were extracted from 12 day-old Col-0 and *mpk3* seedlings and used for the detection of HYL1 proteins by anti-HYL1, while MPK3 was detected by anti-MPK3 antibody. The red dotted line represents the slow migrating HYL1 band. (**B**) The total protein extracted from the *mpk3* seedlings was incubated with (+) or without (-) calf intestinal alkaline phosphatase (CIAP) and further used for the detection of the HYL1 protein. The blue dotted line represents the dephosphorylated HYL1. The plus and minus signs indicate the presence and absence of CIAP. (**C**) A Western blot showing the accumulation of hyper-phosphorylated HYL1 protein in seedlings treated with or without (+/−) protein phosphatase (PPTase) inhibitors. ++ indicates double concentration of PPTase.

**Figure 4 ijms-23-03787-f004:**
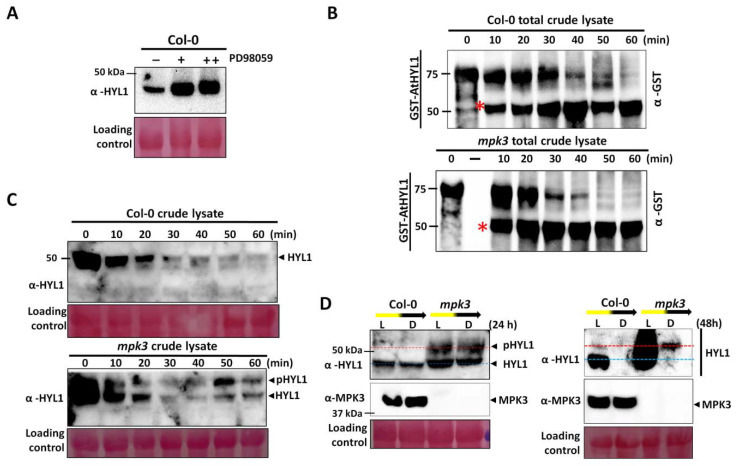
MPK3 promotes AtHYL1 protein degradation. (**A**) Col-0 seedlings (10 days old) were treated with or without (+/−) MAPK inhibitor (PD98059) overnight, and then total protein was used for HYL1 detection. ++ indicates double concentration of PD98059. (**B**) The cell-free degradation assay of the GST-HYL1 protein. Bacterially expressed and purified protein was incubated with the total protein extracted from Col-0 and *mpk3* seedlings for the indicated times, and AtHYL1 protein was detected using anti-GST antibody. The asterisk (*) indicates the proteolytic cleaved product. (**C**) The stability of endogenous HYL1 was monitored by extracting the total protein from Col-0 and *mpk3* seedlings and then incubating at room temperature for the indicated times. An anti-HYL1 antibody was used to detect the HYL1 protein. pHYL1 indicates the phosphorylated protein. (**D**) A Western blot showing the stability of AtHYL1 in Col-0 and *mpk3* seedlings in response to light-to-dark transitions at 24 h and at 48 h. An anti-MPK3 antibody was used to detect the total MPK3 protein. All the above experiments were repeated five times.

**Figure 5 ijms-23-03787-f005:**
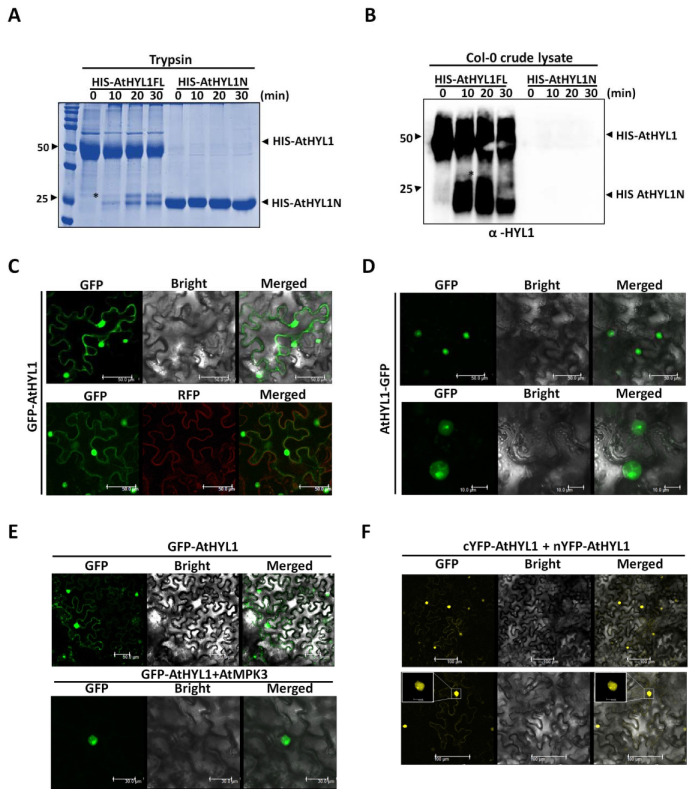
The nucleo–cytosolic shuttling of AtHYL1 is strictly regulated by the co-expression of AtMPK3 and its cytosolic proteolysis. (**A**) A CBB-stained SDS-PAGE showing the cleavage of HIS-AtHYL1 full length and truncated HIS- AtHYL1N by protease trypsin. The proteins were incubated with trypsin for the indicated times, followed by SDS-PAGE, and the gel was stained with CBB. (**B**) An in vitro protein degradation assay of bacterially purified HIS-AtHYL1 full length and truncated HIS-AtHYL1N incubated with crude protein extract (5 µg) from wild type *A. thaliana* (Col-0) for the indicated time periods, followed by immunoblotting with anti-HYL1. The asterisk indicates the cleaved product. The numbers on the left side are the protein molecular weights of the marker. (**C**,**D**) The localization of GFP-AtHYL1 (**C**) and AtHYL1-GFP (**D**) in *N. benthamiana* leaves were monitored under a confocal microscope. (**E**) The localization of GFP-AtHYL1 alone (upper) and with AtMPK3 in *N. benthamiana* leaves. (**F**) A BiFC assay showing the dimerization of cYFP-AtHYL1 and nYFP-AtHYL1 in *N. benthamiana* leaves. Nuclear dimerization is shown in the enlarged subset. The scale bar is shown in each figure section.

## Data Availability

All the nucleotide and protein sequences are available at the TAIR database (https://www.arabidopsis.org/, accessed on 1 June 2016) and the NCBI database (https://www.ncbi.nlm.nih.gov/, accessed on 1 June 2016), and can be accessed with the accession numbers AtHYL1 (AT1G09700), AtMPK3 (AT3G45640), AtSE (AT2G27100), OsDRB1-1 (Os05g24160), OsDRB1-2 (Os11g01869), OsDRB1-3 (Os12g01916), OsDRB1-4 (Os08g29530), and OsMPK3 (Os03g17700.1). The other data underlying this article are available in the article and in its online Appendix A. Additional data related to this article may be shared on reasonable request to the corresponding author.

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
