# Peer review of "Dynamic Phosphorylation of miRNA Biogenesis Factor HYL1 by MPK3 Involving Nuclear–Cytoplasmic Shuttling and Protein Stability in Arabidopsis"

_ijms, 2022, doi:10.3390/ijms23073787_

Round 1

Reviewer 1 Report

The article is devoted to a very interesting and topical topic about the mechanisms of phosphorylation of the HYL1 protein, which is key for the biosynthesis and processing of microRNAs in plants.
The authors did a great and very serious work, in which they almost unambiguously showed the role of AtMPK3 in HYL1 phosphorylation.
The article will be interesting and useful to readers.

Minor remarks need to be corrected

It is necessary to describe the statistical methods used in the article.
The methods must indicate which plant organ was used in the analyses.
It is necessary to indicate the type of lamps and the intensity of light used in the experiments.
If possible, add more articles from 2020 and 2021 to the References.

The name of MS does not fully reflect the content, because in the materials and methods it is written that they used an additional rice plant.

Author Response

Comment 1: It is necessary to describe the statistical methods used in the article.

Response: As suggested we have mentioned the number of times the experiments were repeated to validate the results. The data presented does not warrant any statistics, hence any statistic method is not required.

Comment 2: The methods must indicate which plant organ was used in the analyses.

Response: As suggested we have mentioned the plant organ used for the study in the materials and methods as well as in the figure legends.

Comment 3: It is necessary to indicate the type of lamps and the intensity of light used in the experiments.

Response: As suggested we have mentioned the type of lamp and intensity of light used for the experiments used in the experiments in the materials and methods section.

Comment 4: If possible, add more articles from 2020 and 2021 to the References.

Response: As suggested we have incorporated recent references related to the article in the manuscript and also highlight them in the reference section.

Comment 5:  The name of MS does not fully reflect the content, because in the materials and methods it is written that they used an additional rice plant.

Response: The use of rice OsDRBs, orthologue of Arabidopsis AtHYL1 represented in Fig 2 (D, E and F) is only to substantiate the phosphorylation of Serine at 42nd position in AtHYL1 by MPK3. The manuscript mainly talks about and describes the function of AtHYL1 in Arabidopsis. Based on the above fact the title of the manuscript fully reflects the content.

Reviewer 2 Report

The authors provide new insights onto the regulation of AtHYL1 by AtMPK3 and propose important future perspectives.

The research is well designed and the experiments very well conducted with the correct controls.

The manuscript is well prepared, with comprehensive figures legends.

English language and style are fine.

Author Response

Comments:

The authors provide new insights onto the regulation of AtHYL1 by AtMPK3 and propose important future perspectives.

The research is well designed and the experiments very well conducted with the correct controls.

The manuscript is well prepared, with comprehensive figures legends.

English language and style are fine.

Response:

We thank the Reviewer for the kind words and for appreciating our work.